# Substrate specificity of TOR complex 2 is determined by a ubiquitin-fold domain of the Sin1 subunit

**Hisashi Tatebe[1]\*[†], Shinichi Murayama[1][†], Toshiya Yonekura[1], Tomoyuki Hatano[1], David Richter[2], Tomomi Furuya[1], Saori Kataoka[3], Kyoko Furuita[3], Chojiro Kojima[3,4], Kazuhiro Shiozaki[1,2]\***

[1]Graduate School of Biological Sciences, Nara Institute of Science and Technology, Nara, Japan; [2]Department of Microbiology and Molecular Genetics, University of California, California, United States; [3]Institute for Protein Research, Osaka University, Osaka, Japan; [4]Graduate School of Engineering, Yokohama National University, Yokohama, Japan

**Abstract** The target of rapamycin (TOR) protein kinase forms multi-subunit TOR complex 1 (TORC1) and TOR complex 2 (TORC2), which exhibit distinct substrate specificities. Sin1 is one of the TORC2-specific subunit essential for phosphorylation and activation of certain AGC-family kinases. Here, we show that Sin1 is dispensable for the catalytic activity of TORC2, but its conserved region in the middle (Sin1CRIM) forms a discrete domain that specifically binds the TORC2 substrate kinases. Sin1CRIM fused to a different TORC2 subunit can recruit the TORC2 substrate Gad8 for phosphorylation even in the *sin1* null mutant of fission yeast. The solution structure of Sin1CRIM shows a ubiquitin-like fold with a characteristic acidic loop, which is essential for interaction with the TORC2 substrates. The specific substrate-recognition function is conserved in human Sin1CRIM, which may represent a potential target for novel anticancer drugs that prevent activation of the mTORC2 substrates such as AKT.

**\*For correspondence:** htatebe@ bs.naist.jp (HT); kaz@bs.naist.jp (KS)

[†]These authors contributed equally to this work

**Competing interests:** The authors declare that no competing interests exist.

## Introduction

Sin1 (*SAPK-in*teracting protein 1) was identified in a yeast two-hybrid screen as a protein that interacts with the stress-activated Spc1/Sty1 MAP kinase (MAPK) in the fission yeast *Schizosaccharomyces pombe* (*Wilkinson et al., 1999*). Because of the stress-sensitive phenotype of the *sin1* null (Δ*sin1*) mutant, Sin1 was proposed to regulate the stress MAPK cascade; however, a later scrutiny failed to find a clear functional link between Sin1 and the MAPK cascade (*Ikeda et al., 2008*). Sin1 orthologs have been identified in diverse eukaryotic species through various screens. A partial cDNA clone encoding a human ortholog was isolated in a genetic screen for suppressors of the RAS function in budding yeast (*Colicelli et al., 1991*). Biochemical isolation of proteins binding to mammalian MAPK kinase kinase, MEKK2, also identified a human Sin1 ortholog (*Cheng et al., 2005*). A *Dictyostelium* homolog RIP3 was isolated in a yeast two-hybrid screen for proteins interacting with mammalian Ha-Ras (*Lee et al., 1999*).

Identification of Sin1 orthologs as a component of the TOR (target of rapamycin) protein complex provided an important clue for the cellular function of Sin1 (*Loewith et al., 2002*; *Wedaman et al., 2003*; *Lee et al., 2005*; *Frias et al., 2006*; *Jacinto et al., 2006*; *Yang et al., 2006*). TOR is a serine/threonine-specific protein kinase conserved from yeast to humans, forming two distinct protein complexes referred to as TOR complex 1 (TORC1) and TOR complex 2 (TORC2), the latter of which contains Sin1 as a stable subunit (*Wullschleger et al., 2006*). Mammalian TORC2 (mTORC2) functions

**eLife digest** Human cells contain a group of proteins that together make up a molecular machine called the TOR complex 2, or TORC2 for short. TORC2 contains an enzyme that changes the activities of other proteins by tagging them with phosphate groups, a process known as phosphorylation. TORC2 has control over many proteins and it can significantly affect the overall behaviour of a cell. Yet it is not known how TORC2 specifically targets these proteins over others in the cell.

The components of TORC2 are very similar in yeast and humans, and Tatebe, Murayama et al. have now used a species of yeast called *Schizosaccharomyces pombe* to examine the proteins in TORC2. The experiments revealed that one protein called Sin1 is responsible for selecting the targets for TORC2. Analysis of Sin1 from both yeast and humans identified that the ability to select target proteins is due to a specific part in the middle of this protein. Further experiments revealed that the middle portion of Sin1 has a shape known as a ubiquitin-fold, which is a common feature of proteins that selectively interact with other proteins.

More detailed examination of the middle part of Sin1 is still needed to understand why it attaches to some proteins and not others. In particular, TORC2 is known to control a protein called AKT, which plays an important role in some cancers. Understanding how to specifically stop TORC2 from activating AKT could result in new and targeted approaches to treating these cancers.

downstream of the PI3K pathway, activating AGC-family protein kinases, such as AKT, protein kinase Cα (PKCα) and SGK1, by direct phosphorylation of their hydrophobic motif (*Sarbassov et al., 2004*; *Hresko and Mueckler, 2005*; *Sarbassov et al., 2005*; *Guertin et al., 2006*; *Facchinetti et al., 2008*; *García-Martínez and Alessi, 2008*; *Ikenoue et al., 2008*; *Lu et al., 2010*). Thus, mTORC2 appears to serve as a signaling node that controls protein kinases pivotal to the regulation of cellular metabolism and proliferation (*Cybulski and Hall, 2009*). In the SIN1 knockout/knockdown cell lines, no functional mTORC2 is formed (*Frias et al., 2006*; *Jacinto et al., 2006*; *Yang et al., 2006*) and therefore, it has been proposed that SIN1 is required for the assembly/integrity of mTORC2. In addition, immunoprecipitation of SIN1 resulted in co-purification of AKT, raising the possibility that SIN1 might serve as a scaffold for AKT (*Jacinto et al., 2006*; *Cameron et al., 2011*).

Also in fission yeast, Sin1 is a stable subunit of TORC2 that assembles around Tor1, one of the two TOR kinases in this organism (*Hayashi et al., 2007*; *Matsuo et al., 2007*). Fission yeast TORC2 is regulated by the Rab GTPase Ryh1 (*Tatebe et al., 2010*) and phosphorylates S546 within the hydrophobic motif of the Gad8 kinase, which has significant sequence homology to human AKT and SGK1 (*Matsuo et al., 2003*; *Ikeda et al., 2008*). As in the *tor1* null (Δ*tor1*) strain, Gad8-S546 is not phosphorylated in the Δ*sin1* strain (*Ikeda et al., 2008*; *Tatebe et al., 2010*). In addition, Δ*sin1* mutant cells show phenotypes indistinguishable from those of Δ*tor1* and Δ*gad8* cells, including sterility and hypersensitivity to stress (*Wilkinson et al., 1999*; *Kawai et al., 2001*; *Weisman and Choder, 2001*; *Matsuo et al., 2003*; *Ikeda et al., 2008*). These observations indicate that Sin1 plays a critical role as a TORC2 subunit in phosphorylating and activating Gad8, although the exact molecular function of Sin1 remains obscure.

In this report, we present a set of evidence that the Sin1 subunit of fission yeast TORC2 binds and recruits Gad8 kinase for phosphorylation. The evolutionarily conserved central region of Sin1, called conserved region in the middle (CRIM) (*Schroder et al., 2004*), is sufficient to recognize and bind Gad8. In addition, the CRIM domain of human SIN1 can bind mTORC2 substrates such as AKT, PKCα and SGK1, suggesting the conserved role of Sin1 CRIM in substrate recognition by TORC2. The NMR structure of Sin1 CRIM shows a ubiquitin-like fold with a negatively charged, protruding loop, which plays a key role in recruiting the TORC2 substrates.

# Results

## Sin1 is not required for the integrity of TORC2 in fission yeast

In mTORC2, it has been proposed that the SIN1 subunit is required for the interaction between mTOR and the RICTOR subunit, as their association was not detectable in the *SIN1* knockout cell line (*Jacinto et al., 2006*). Although this observation might be due to the reduced level of RICTOR in the absence of SIN1 (*Frias et al., 2006*), we tested whether absence of Sin1 affects the TORC2 integrity in the fission yeast *S. pombe*, in which TORC2 is not essential for cell viability. Immunoprecipitation of the FLAG epitope-tagged Tor1 detected its association with the RICTOR ortholog Ste20 in wild-type, Δ*sin1*, Δ*wat1* and Δ*bit61* cell lysate, indicating that Tor1 interacts with Ste20 in the absence of Sin1 or other TORC2 subunits (*Figure 1A*). Reciprocal experiments also confirmed this conclusion (*Figure 1B*). Similarly, the mLST8 ortholog Wat1 co-precipitated with Tor1 from the lysate of strains lacking other TORC2 subunits, including Sin1 (*Figure 1—figure supplement 1A and B*). It has also been found that Sin1 is not required for the association of Bit61, a fission yeast ortholog of PROTOR/PRR5 (*Hayashi et al., 2007*), with TORC2 (*Tatebe and Shiozaki, 2010*).

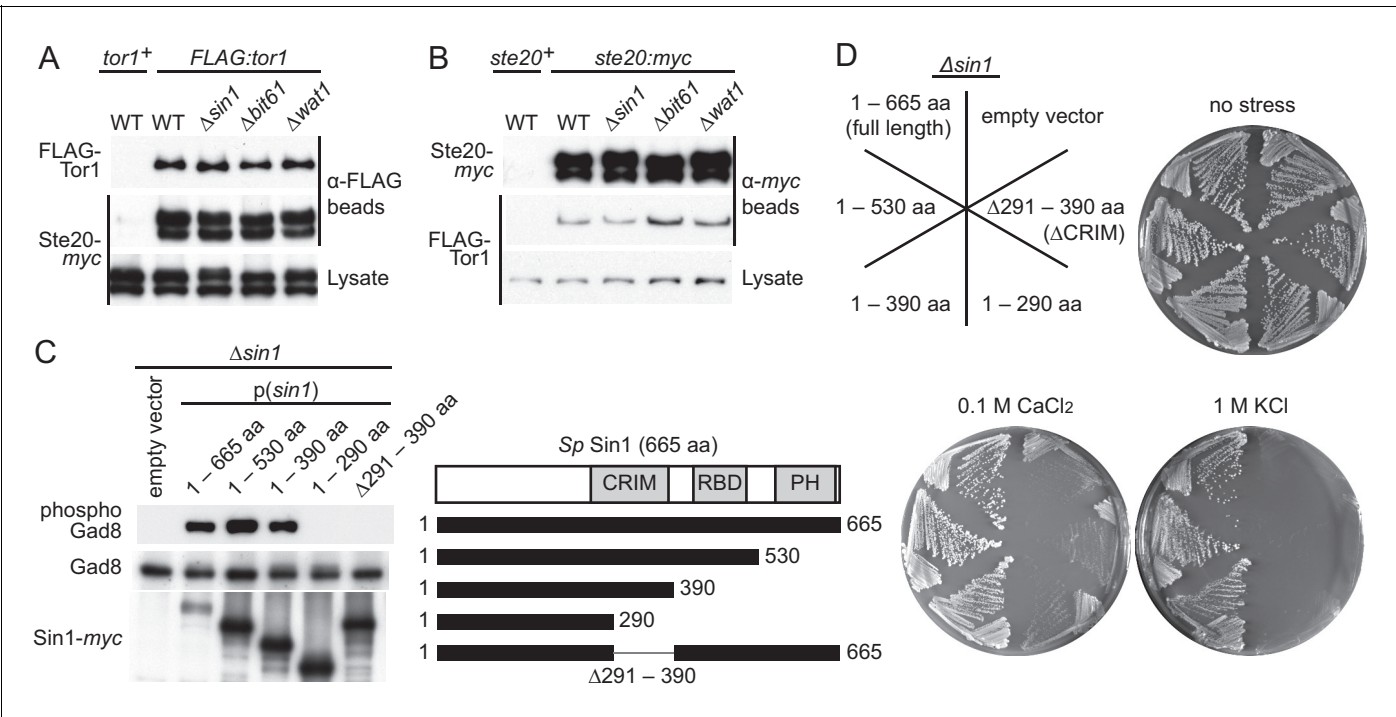

**Figure 1.** CRIM, but not the RBD and PH domain, is essential for Sin1 function. (A, B) The RICTOR ortholog Ste20 associates with Tor1 in the absence of Sin1. (A) Co-purification of Ste20-*myc* with FLAG-Tor1 was tested by anti-FLAG ('α-FLAG') affinity beads in wild-type (CA7087), Δ*sin1* (CA7143), Δ*bit61* (CA7150) and Δ*wat1* (CA7151) cell lysate. A *ste20:myc* strain expressing untagged Tor1 (CA6435) was used as a negative control (*tor1*[+]). (B) Co-purification of FLAG-Tor1 with Ste20-*myc* was tested by anti-myc ('α-myc') affinity beads as in (A). A *FLAG:tor1* strain expressing untagged Ste20 (CA6530) was used as a negative control (*ste20*[+]). (C) The CRIM, but not the RBD and C-terminal PH domain, of Sin1 is required for TORC2 to phosphorylate Gad8. TORC2-dependent phosphorylation of Gad8-S546 and total Gad8 were detected by immunoblotting (*Tatebe et al., 2010*) in cell lysate from a Δ*sin1* strain (CA5126) carrying plasmids to express full-length (1–665 aa) or various Sin1 fragments fused to the *myc* epitope. (D) Expression of the mutant Sin1 lacking the C-terminus rescues the Δ*sin1* phenotype. Their stress sensitivity was evaluated by growth on solid YES medium containing either 1 M KCl or 0.1 M CaCl2 for 2 days at 30°C. CRIM, Conserved region in the middle; PH, Pleckstrin homology; RBD, Ras-binding domain.

The following figure supplements are available for figure 1:

**Figure supplement 1.** Sin1 is dispensable for the interaction between Tor1 and the LST8 ortholog Wat1.

**Figure supplement 2.** The CRIM domain is not required for Sin1 to bind the other TORC2 subunits.

Together, these results indicate that the Sin1 subunit is not crucial for the integrity of *S. pombe* TORC2 and, even in the absence of Sin1, the other TORC2 subunits assemble into a complex.

## The CRIM domain, but not the RBD and PH domain, is essential for the Sin1 function

Sin1 orthologs from yeast to humans have a Ras-binding domain (RBD) and a pleckstrin homology (PH) domain at their C terminus (*Figure 1C*) (*Schroder et al., 2007*; *Pan and Matsuura, 2012*). It was previously reported that the C-terminal 164 amino acid residues of *S. pombe* Sin1, including the PH domain and a part of the RBD, are essential for Sin1 function in vivo (*Wilkinson et al., 1999*). Unexpectedly, however, our immunoblotting experiments with the antibodies against Gad8 phosphorylated at S546 within the hydrophobic motif (*Tatebe et al., 2010*) found that the TORC2-dependent phosphorylation of Gad8 was not compromised in *S. pombe* cells expressing Sin1 lacking the RBD and PH domain (1–530 aa and 1–390 aa, *Figure 1C*). Consistently, the stress-sensitive phenotypes of the Δ*sin1* mutant (*Matsuo et al., 2007*; *Ikeda et al., 2008*) was complemented by expressing those truncated Sin1 proteins (*Figure 1D*), confirming that the C-terminal region of Sin1 including its PH domain is dispensable for the Sin1 function in fission yeast. On the other hand, further C-terminal truncation (1–290 aa) or internal deletion (Δ291–390 aa), which removes the CRIM (*Schroder et al., 2004*), completely abolished the Sin1 function (*Figure 1C and D*), although those Sin1 mutant proteins were still capable of forming a complex with the other TORC2 subunits (*Figure 1—figure supplement 2*). These results strongly suggest an essential role of the Sin1 CRIM in the TORC2 function.

## Sin1 CRIM domain binds the AGC kinases that are phosphorylated by TORC2

During a yeast two-hybrid screen to identify fission yeast proteins that interact with the Gad8 kinase (Materials and methods), we isolated cDNA clones encoding Sin1. Immunoprecipitation of FLAG epitope-tagged Gad8 from fission yeast cell lysate co-purified a small amount of Sin1 (*Figure 2A*), confirming weak but detectable interaction between the two proteins in fission yeast. Yeast two-hybrid assays using a series of truncated Sin1 fragments indicated that Gad8 binds to the amino acid residues 281–400 of Sin1 (*Figure 2—figure supplement 1A*), which mostly overlap with the CRIM domain and are thus referred to as SpSin1CRIM hereafter. When SpSin1CRIM fused to GST was expressed in *S. pombe*, Gad8 was co-purified with this GST fusion (*Figure 2B*), confirming that this 120-residue region of Sin1 is sufficient for the interaction with Gad8. Moreover, induced overexpression of the GST-SpSin1CRIM fusion significantly reduced the Gad8-S546 phosphorylation (*Figure 2C*); one likely possibility is that the expressed SpSin1CRIM fragment can compete with TORC2 for Gad8.

Because of the evolutionary conservation of the CRIM domain among Sin1 orthologs (*Schroder et al., 2004*), we examined whether human SIN1 CRIM binds to the mTORC2 substrates, such as AKT, PKCα and SGK1 (*Sarbassov et al., 2005*; *Jacinto et al., 2006*; *García-Martínez and Alessi, 2008*; *Cameron et al., 2011*). Although the physical interaction between human SIN1 and AKT has been controversial (*Jacinto et al., 2006*; *Cameron et al., 2011*; *Lu et al., 2011*), our yeast two-hybrid assays found that the residues 122–314 of human SIN1, referred to as HsSin1CRIM hereafter, interacted with AKT as well as PKCα (*Figure 2—figure supplements 1C,D* and *2*). Furthermore, in an in vitro binding assay, AKT, PKCα as well as SGK1 were co-precipitated with bacterially produced HsSin1CRIM (*Figure 2D*). Although the interaction between SIN1 and SGK1 has been reported (*Lu et al., 2011*), our experiments showed for the first time that the CRIM domain of SIN1 is sufficient for the interaction with SGK1. Another mTORC2 substrate, PKCε, is known to interact with SIN1 fragments that include the CRIM domain (*Cameron et al., 2011*). Together, these results strongly suggest that the ability of the CRIM domain to bind the TORC2 substrate kinases is conserved between *S. pombe* and humans.

Despite the structural similarity among the AGC kinase family members, TORC2 phosphorylates only a subset of them; human S6K1 and its fission yeast ortholog Psk1 are AGC kinases whose hydrophobic motif is phosphorylated by TORC1 rather than TORC2 (*Magnuson et al., 2012*; *Nakashima et al., 2012*). Consistently, human S6K1 and *S. pombe* Psk1 showed no detectable binding to HsSin1CRIM and SpSin1CRIM, respectively (*Figure 2D and E*). Psk1 and other *S. pombe*

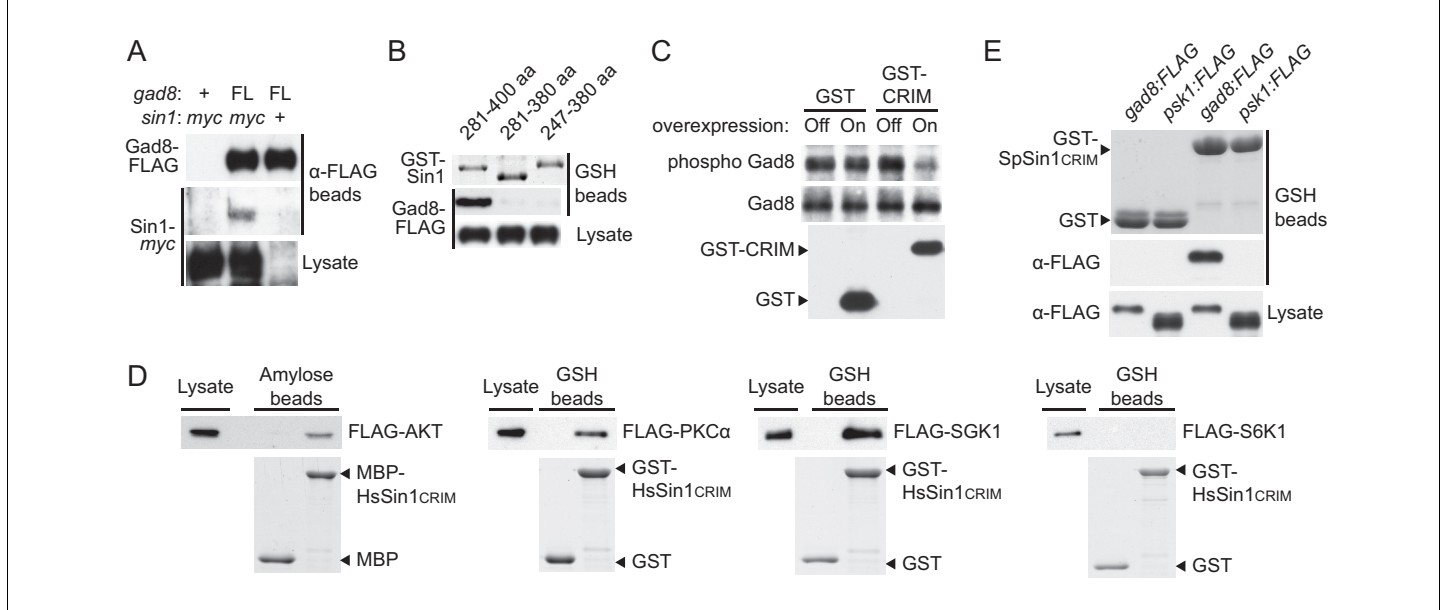

**Figure 2.** Sin1CRIM binds specifically to the TORC2 target AGC kinases. (**A**) Sin1 interacts with the TORC2 substrate Gad8 in *S. pombe*. Co-purification of Sin1-*myc* with Gad8-FLAG ('FL') precipitated by anti-FLAG beads was tested in a *sin1:myc gad8:FLAG* strain (lane 2; CA6993). *sin1:myc gad8+* (lane 1; CA6984) and *gad8:FLAG sin1+* (lane 3; CA6281) strains were used as negative controls. (**B**) SpSin1CRIM can bind Gad8 in *S. pombe*. SpSin1CRIM fused to GST was expressed in a *gad8:FLAG* strain (CA6281), and proteins collected on glutathione (GSH)-beads and the cell lysate were analyzed by Coomassie blue staining and anti-FLAG immunoblotting. (**C**) Overexpressed SpSin1CRIM inhibits Gad8 phosphorylation by TORC2. GST or GST-SpSin1CRIM (247–400) were induced ('On') from the thiamine-repressible *nmt1* promoter and the crude cell lysate was analyzed by immunoblotting. (**D**) Human Sin1CRIM binds mTORC2 substrates, but not the mTORC1 substrate S6K1. FLAG-tagged AKT, PKCα, Sgk1 and S6K1 were expressed in HEK-293T and the cell lysate was incubated with bacterially produced MBP- or GST-fused HsSin1CRIM, which were immobilized onto amylose- and GSH-beads, respectively. Proteins bound to the beads and the cell lysate were analyzed by anti-FLAG immunoblotting and Coomassie blue staining. (**E**) SpSin1CRIM does not interact with Psk1, a *S. pombe* TORC1 substrate. Bacterially produced GST-SpSin1CRIM bound to GSH-beads was incubated with cell lysate from *gad8:FLAG* (CA6281) and *psk1:FLAG* (CA8070) strains of *S. pombe*. Proteins bound to the GSH-beads and the cell lysate were analyzed by anti-FLAG immunoblotting and Coomassie blue staining.

The following figure supplements are available for figure 2:

**Figure supplement 1.** Interaction between fission yeast Sin1 – Gad8 and human Sin1 – AKT2 in yeast two-hybrid assays.

**Figure supplement 2.** Yeast two-hybrid assay for interaction between HsSin1CRIM and PKCα.

**Figure supplement 3.** Yeast two-hybrid assay for interaction between SpSin1CRIM and fission yeast Sck1, Sck2, Psk1, Pck1, Pck2.

**Figure supplement 4.** Yeast two-hybrid assay for cross-species interaction between Sin1, Gad8, and AKT2.

AGC kinases that are not regulated by TORC2, including Sck1, Sck2, Pck1 and Pck2 (*Nakashima et al., 2012*; *Madrid et al., 2015*), also failed to interact with SpSin1CRIM in yeast two-hybrid assays (*Figure 2—figure supplement 3*). Thus, it appears that, in both fission yeast and humans, the CRIM domain of Sin1 can selectively bind the TORC2 substrates among the members of the AGC kinase family. It should be noted, however, that the CRIM domain does not seem to recognize the TORC2 phosphorylation sites per se, because CRIM can interact with Gad8 and AKT lacking their C-terminal hydrophobic motif (*Figure 2—figure supplement 1B and D*).

## Sin1 CRIM has a protruding acidic loop involved in the recruitment of TORC2 substrates

For better understanding of the molecular mechanism by which the CRIM domain recognizes the TORC2 substrates, we sought amino acid residues essential for the SpSin1CRIM function. Random mutagenesis followed by yeast two-hybrid screens isolated several SpSin1CRIM missense mutations

that abolish the interaction with Gad8 (*Figure 3A*). Each of the isolated mutations was introduced into GST-SpSin1CRIM, and all the successfully expressed mutant proteins showed significantly compromised interaction with Gad8 in fission yeast cells (*Figure 3B*; data not shown). Those confirmed mutations were then introduced to the full-length Sin1 protein tagged with the *myc* epitope and expressed in the Δ*sin1* strain for further analyses. Complementation of the Δ*sin1* defects in Gad8 phosphorylation (*Figure 3C*) and cellular stress resistance (*Figure 3D*) was not observed even with the mutant Sin1 proteins whose expression was comparable to that of the wild-type protein. On the

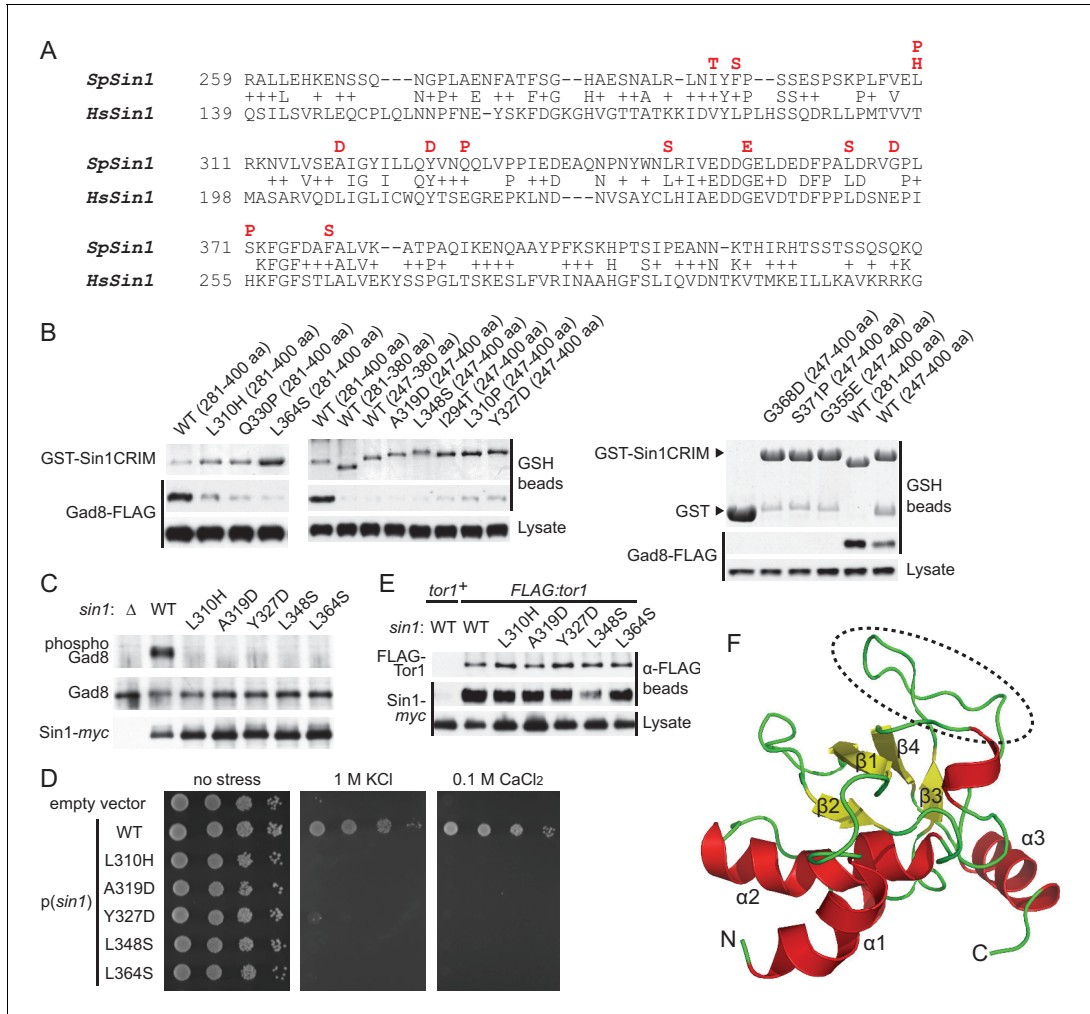

**Figure 3.** Sin1CRIM forms a ubiquitin-fold domain that binds TORC2 substrates. (A) Yeast two-hybrid screens isolated SpSin1CRIM mutations that abrogate the interaction with Gad8 (red). Pairwise alignment of the fission yeast and human CRIM sequences was performed using the SSEARCH program (*Pearson, 1991*). (B) Defective interaction of the mutant SpSin1CRIM with Gad8. The Sin1CRIM (residues 247–400 or 281–400) fragments and those with the indicated mutations were expressed as GST-fusion in a *gad8:FLAG* strain of *S. pombe* (CA6281), and proteins collected on GSH-beads and the cell lysate were detected by Coomassie blue staining and anti-FLAG immunoblotting. (C) TORC2 with the Sin1CRIM mutations fails to phosphorylate Gad8-S546. A Δ*sin1* strain (CA5126) was transformed with a plasmid to express *myc*-tagged wild-type or mutant Sin1 carrying one of the CRIM mutations, and the cell lysate was analyzed by immunoblotting. (D) Stress sensitivity of the Sin1 mutants in (C) was examined by growth on YES agar medium containing either 1 M KCl or 0.1 M CaCl2. (E) CRIM mutations do not compromise the incorporation of Sin1 into TORC2. A Δ*sin1 FLAG: tor1* strain (CA6870) was transformed with the plasmids used in (C), and co-purification of the wild-type and mutant Sin1-*myc* with FLAG-Tor1 precipitated by anti-FLAG beads was tested. A Δ*sin1* strain expressing untagged Tor1 (CA5126) was used as a negative control (*tor1*[+]). (F) A ribbon representation of the lowest energy structure of SpSin1CRIM (272–397). The protruding acidic loop region is marked by a dashed oval.

The following figure supplement is available for figure 3:

**Figure supplement 1.** Mutated residues of SpSin1CRIM in the tertiary structure.

other hand, with the exception of the marginal effect of L348S, those amino acid substitutions within CRIM had no apparent impact on the ability of Sin1 to interact with Tor1 (*Figure 3E*). Thus, these Sin1 mutant proteins were incorporated into TORC2, but the mutations to the CRIM domain compromised the interaction with Gad8, leading to the inability of TORC2 to phosphorylate Gad8.

The experiment above demonstrated that the CRIM domain of Sin1 is essential for TORC2 to bind and phosphorylate the substrate Gad8. However, the isolated mutations are scattered throughout the SpSin1CRIM (*Figure 3A*), providing little mechanistic insight into the substrate recognition. NMR chemical shifts and $^1$H-$^{15}$N heteronuclear NOE values (*Kataoka et al., 2014*; *Furuita et al., 2014*) indicated that SpSin1CRIM is a highly loop-rich but structurally ordered. In general, loop-rich proteins are difficult targets to determine the structures by X-ray and NMR. Therefore, a newly developed method that allows determination of loop-rich protein structures by paramagnetic relaxation enhancement NMR techniques (*Furuita et al., 2014*) was applied to SpSin1CRIM, with additional structure refinement by the XPLOR-NIH software. The solution structure of SpSin1CRIM has three α helices and four β strands, which are arranged around the longest helix α2 (*Figure 3F* and *Table 1*; the atomic coordinates of the refined structure have been submitted to the Protein Data Bank). The overall structure of SpSin1CRIM belongs to the ubiquitin superfold family (*Kiel and Serrano, 2006*), but more than half of the structured region consists of loops.

The majority of the mutations that abrogate the Gad8 binding (*Figure 3A*) are mapped to the buried residues in the structure of SpSin1CRIM (*Figure 3—figure supplement 1*) and are likely to significantly perturb the overall folding of the domain, thus disrupting the functional TORC2-Gad8 interaction (*Figure 3C*). A distinctive characteristic of SpSin1CRIM is its protruding loop structure formed around residues 352–361 (*Figures 3F* and *4A*), which are rich in acidic amino acid residues with solvent-exposed side chains (*Figure 4—figure supplement 1*). In addition, the primary sequence of this loop region is highly conserved among Sin1 orthologs (*Figure 4B* and *Figure 4—figure supplement 2*). Substitutions of the seven acidic residues in this loop region with asparagine and glutamine ('poly NQ', *Figure 4B*) completely eliminated the affinity of SpSin1CRIM for Gad8 (*Figure 4C*) without significantly disturbing the overall folding (*Figure 4—figure supplement 3*), suggesting that the negative charge of the protruding loop is essential. Further analysis of individual substitution mutants suggested that the acidic residues located in the second half of the loop (D358, E359 and D360) play more important roles (*Figure 4—figure supplement 4*). Two conserved hydrophobic residues within this loop region, L357 and F361, also appear to contribute to the interaction with Gad8; alanine substitution of each of these residues partially compromised the Gad8 binding (*Figure 4—figure supplement 5*), and the L357A/F361A double mutant ('AA', *Figure 4B*) showed very little interaction with Gad8 (*Figure 4C*). Consistently, the TORC2-dependent phosphorylation of Gad8-S546 was dramatically impaired in strains expressing Sin1 with those amino-acid substitutions (*Figure 4D*), which did not affect the association between Sin1 and the Tor1 kinase within the TORC2 complex (*Figure 4E*). These results indicate that the protruding acidic loop of SpSin1CRIM is essential for TORC2 to recruit and phosphorylate its substrate, Gad8.

The corresponding 'poly NQ' and 'AA' substitutions in the HsSin1CRIM (*Figure 4B*) partially compromised its affinity for AKT (*Figure 4F*). The importance of the conserved acidic residues was further tested by expressing the 'poly NQ' SIN1 mutant protein in a human cell line where the *SIN1* gene was inactivated by CRISPR/Cas9-mediated genome editing (*Cong et al., 2013*; *Mali et al., 2013*) (*Figure 4—figure supplement 6*). As reported previously (*Frias et al., 2006*; *Jacinto et al., 2006*; *Yang et al., 2006*), absence of functional SIN1 resulted in the loss of mTORC2-dependent phosphorylation of AKT-S473 in the hydrophobic motif, a defect rescued by expressing wild-type SIN1 (*Figure 4G*). The AKT phosphorylation was hardly detectable when re-introduced SIN1 lacks the entire CRIM domain ('ΔCRIM') or its acidic loop region of residues 236–245 ('Δacidic') (*Figure 4G*), although these mutant SIN1 proteins can interact with mTOR and RICTOR (*Figure 4—figure supplement 7*). Only a low level of the AKT-S473 phosphorylation was observed when the 'poly NQ' mutant was expressed (*Figure 4G*), and similar results were obtained for the mTORC2-dependent phosphorylation of AKT-T450 in the turn motif and PKCα-S657 in the hydrophobic motif (*Figure 4—figure supplement 8*). Together, these observations strongly suggest that the conserved acidic residues within the CRIM domain are important for the SIN1 function as mTORC2 subunit.

**Table 1.** Structural statistics for SpSin1CRIM. RDC correlation coefficient was caluculated using the program PALES (*Zweckstetter and Bax, 2000*). Ramachandran analysis was performed using PRO-CHECK 3.5.4 (*Laskowski et al., 1996*). Deviations from ideal geometry was caluculated using PDB Validation Server.

| NOE upper distance restraints | |
|---|---|
| Short-range ($|i-j|<=1$) | 636 |
| Medium-range ($1<|i-j|<5$) | 132 |
| Long-range ($5<=|i-j|$) | 199 |
| Total | 967 |
| PRE distance restraints | 748 |
| Dihedral angle restraints | |
| $\varphi$ | 99 |
| $\psi$ | 110 |
| $\chi_1$ | 12 |
| Hydrogen-bond restraints | 0 |
| RMS Deviations (272–397) (Å) | |
| backbone | $0.89 \pm 0.11$ |
| heavy | $1.36 \pm 0.12$ |
| RDC correlation coefficient | $0.94 \pm 0.02$ |
| Violations | |
| NOE (>0.5 Å) | 0 |
| PRE (>0.5 Å) | 0 |
| Dihedral (>5°) | 0 |
| Maximum violation | |
| NOE (Å) | 0.49 |
| PRE (Å) | 0.15 |
| Dihedral (°) | 3.1 |
| Ramachandran analysis (272–397) (%) | |
| Most favored regions | 91.3 |
| Additional allowed regions | 7.7 |
| Generously allowed regions | 0.5 |
| Disallowed regions | 0.6 |
| Deviations from ideal geometry | |
| Bond lingths (Å) | 0.011 |
| Bond angles (°) | 0.14 |

## CRIM-dependent substrate recruitment determines the specificity of TORC2

Although SpSin1CRIM alone cannot replace the Sin1 function in vivo (*Figure 5—figure supplement 1*), we found that it complemented the stress-sensitive phenotypes of the Δ*sin1* mutant when fused to Ste20, the RICTOR equivalent in *S. pombe* TORC2 (*Figure 5A*). Consistently, expression of the Ste20-SpSin1CRIM fusion was found to induce phosphorylation of Gad8-S546 in the Δ*sin1* strain (*Figure 5B*). With the L348S or L364S substitutions that prevent SpSin1CRIM from binding Gad8 (*Figure 3B*), the Ste20-SpSin1CRIM failed to bring about the Gad8 phosphorylation (*Figure 5B*). As expected, these substitutions as well as the 'poly NQ' and 'AA' mutations to the acidic loop (*Figure 4*) abrogated the ability of Ste20-SpSin1CRIM to complement the Δ*ste20* Δ*sin1* phenotype (*Figure 5—figure supplement 2*). Therefore, the hybrid subunit enables phosphorylation and activation of Gad8

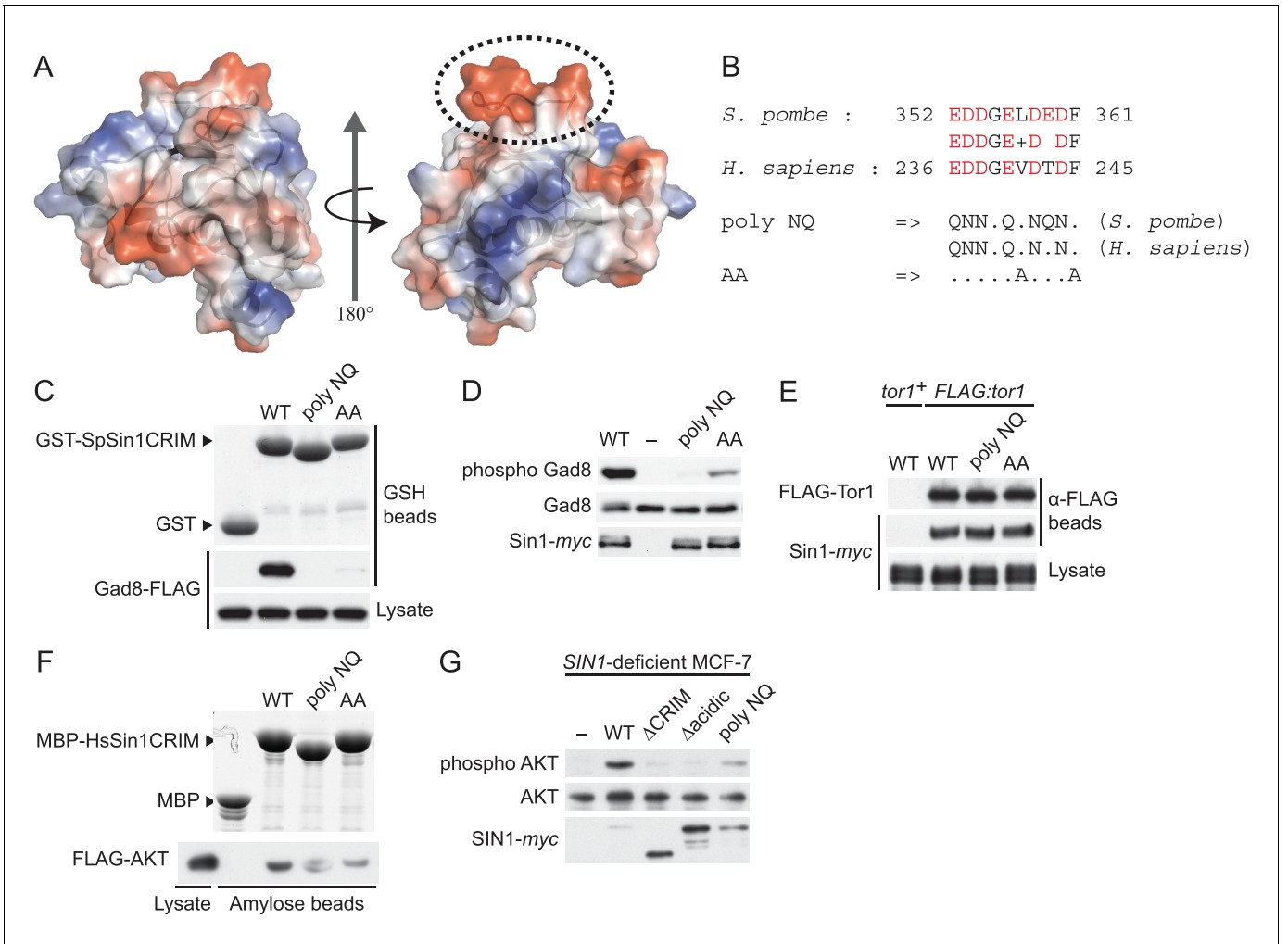

**Figure 4.** The protruding acidic loop of CRIM is essential for Sin1 to interact with TORC2 target kinases. (**A**) The molecular surface of SpSin1CRIM colored according to the electrostatic potential ranging from positive (blue) to negative (red) charge. The protruding acidic loop is marked by a dashed oval. (**B**) Alignment of the fission yeast and human Sin1CRIM sequences that correspond to the acidic protrusion. The 'poly NQ' and 'AA' show the amino acid substitutions used in (**C**) ~ (**G**). (**C**) The acidic and hydrophobic residues in the protruding loop of SpSin1CRIM are essential for Gad8 binding in vitro. The recombinant wild-type and mutant GST-SpSin1CRIM on GSH-beads were incubated with cell lysate from a *gad8:FLAG* strain (CA6281). Proteins bound to the beads and the cell lysate were analyzed by anti-FLAG immunoblotting and Coomassie blue staining. (**D**) TORC2 with the mutations to the acidic loop of Sin1CRIM fails to phosphorylate Gad8. A Δ*sin1* strain (CA5126) was transformed with a plasmid to express *myc*-tagged wild-type or mutant Sin1 carrying the poly-NQ and AA mutations, and the cell lysate was analyzed by immunoblotting. (**E**) The acidic loop mutations do not compromise the incorporation of Sin1 into TORC2. A Δ*sin1 FLAG:tor1* strain (CA6870) was transformed with the plasmids used in (**D**), and co-purification of the wild-type and mutant Sin1-*myc* with FLAG-Tor1 precipitated by anti-FLAG beads was tested. A Δ*sin1* strain expressing untagged Tor1 (CA5126) was used as a negative control (*tor1*[+]). (**F**) The acidic and hydrophobic residues in the conserved loop region of HsSin1CRIM are important to bind AKT. FLAG-tagged AKT was expressed in HEK-293T, and the cell lysate was incubated with the recombinant wild-type and mutant MBP-HsSin1CRIM on amylose-beads. Proteins bound to the beads and the cell lysate were analyzed as in (**C**). (**G**) The conserved acidic loop region of HsSin1CRIM is important for mTORC2-dependent phosphorylation of AKT-S473. The wild-type and mutant SIN1 lacking the entire CRIM ('ΔCRIM') or acidic region ('Δacidic') as well as the poly NQ SIN1mutant were expressed with the *myc* tag in the *SIN1*-deficient MCF-7 cells. Insulin-stimulated phosphorylation of AKT-S473 was analyzed by immunoblotting.

The following figure supplements are available for figure 4:

**Figure supplement 1.** The acidic side chains of the conserved loop region do not interact with other residues of the CRIM domain.

**Figure supplement 2.** The acidic loop region is highly conserved among Sin1 orthologs.

**Figure supplement 3.** An overlay of [1]H-[15]N HSQC spectra of the wild-type (blue) and 'poly NQ' mutant (red) of SpSin1CRIM.

*Figure 4 continued on next page*

*Figure 4 continued*

**Figure supplement 4.** Point mutations to the acidic residues in the conserved SpSin1CRIM loop compromise the interaction with Gad8.

**Figure supplement 5.** Alanine substitution of the conserved hydrophobic residues within the SpSin1CRIM loop reduces the Gad8 binding.

**Figure supplement 6.** Inactivation of *SIN1* by CRISPR/Cas9-mediated genome editing was confirmed by immunoblotting and genomic sequencing.

**Figure supplement 7.** The CRIM domain of human Sin1 is not required for its interaction with mTOR and RICTOR to form mTORC2.

**Figure supplement 8.** The human Sin1CRIM and its acidic loop are essential for phosphorylation of the mTORC2 targets.

in the absence of Sin1 most likely by recruiting Gad8 to TORC2. Thus, the CRIM domain alone appears to be sufficient for the substrate-recruiting function of the Sin1 subunit.

To further confirm that Sin1 is not essential for the catalytic activity of TORC2 per se, we expressed Gad8 as a fusion to the Ste20 subunit, so that Gad8 physically associates with TORC2 even in the absence of Sin1. As shown in *Figure 5C*, the Gad8 phosphorylation was detectable within the Ste20-Gad8 fusion protein in *Δsin1* cells that express the wild-type Tor1 kinase, but hardly in those expressing Tor1-D2137A with compromised catalytic activity (*Tatebe et al., 2010*). Thus, TORC2 lacking Sin1 can phosphorylate the hydrophobic motif of Gad8 recruited to TORC2, indicating that Sin1 is not essential for the catalytic activity of TORC2.

We also constructed a fusion gene between *gad8*+ and *mip1*+, which encodes a *S. pombe* TORC1 subunit homologous to mammalian RAPTOR (*Matsuo et al., 2007*; *Alvarez and Moreno, 2006*), and expressed it in the *Δsin1* mutant. The hydrophobic motif of Gad8 within the Mip1-Gad8 fusion was phosphorylated (*Figures 5D* and 0 min), which became negligible immediately after nitrogen starvation that inactivates TORC1 but not TORC2 (*Alvarez and Moreno, 2006*; *Uritani et al., 2006*; *Matsuo et al., 2007*; *Nakashima et al., 2012*; *Hatano et al., 2015*). This result suggests that TORC1 can phosphorylate Gad8, a TORC2 substrate, when Gad8 is artificially brought to TORC1. Indeed, co-precipitation experiments showed that the Mip1-Gad8 fusion protein forms a complex preferentially with Tor2, the catalytic subunit of TORC1, rather than with Tor1 (*Figure 5E*). This association with Tor2 was significantly compromised by introducing alanine substitutions to residues 316–318 or 366–369 of Mip1 in the fusion protein ('DLF->AAA' and 'NWIF->AAAA', respectively); the equivalent substitutions in human RAPTOR are known to abrogate its interaction with mTOR (*Kim et al., 2002*). As expected, the compromised interaction of Mip1-Gad8 with Tor2 dramatically undermined the phosphorylation of the Gad8 hydrophobic motif (*Figure 5F*), confirming that the detected phosphorylation is dependent on the association of Gad8 with TORC1. Although TORC1 and TORC2 in fission yeast have different catalytic subunits, Tor2 and Tor1, respectively (*Hayashi et al., 2007*; *Matsuo et al., 2007*), TORC1 is capable of phosphorylating the TORC2 substrate Gad8, when Gad8 physically interacts with TORC1.

## Discussion

Unlike in other model organisms, TORC2 in the fission yeast *S. pombe* is not essential for viability. In this study, we utilized this experimental organism to clarify the molecular function of Sin1, an evolutionarily conserved subunit of TORC2.

In the *sin1* null mutant of *S. pombe*, we observed assembly of the Sin1-free TORC2, which can phosphorylate Gad8 fused to the Ste20 subunit, suggesting that ensuring the functional assembly and integrity of TORC2 is not an intrinsic function of Sin1. Although the budding yeast Sin1 ortholog was implicated in targeting TORC2 to the plasma membrane (*Berchtold and Walther, 2009*), the membrane localization of fission yeast TORC2 is not dependent on Sin1 (*Tatebe et al., 2010*).

On the other hand, our data presented here demonstrate that Sin1 is the substrate-recruiting subunit of *S. pombe* TORC2. Furthermore, the critical role of the CRIM domain in substrate recognition appears to be conserved in the human SIN1. We showed for the first time that recombinant HsSin1CRIM can bind the mTORC2 substrates such as AKT, PKCα and SGK1 in vitro, but not the TORC1 substrate S6K1. Interestingly, the fission yeast CRIM also interacts with AKT in yeast two-

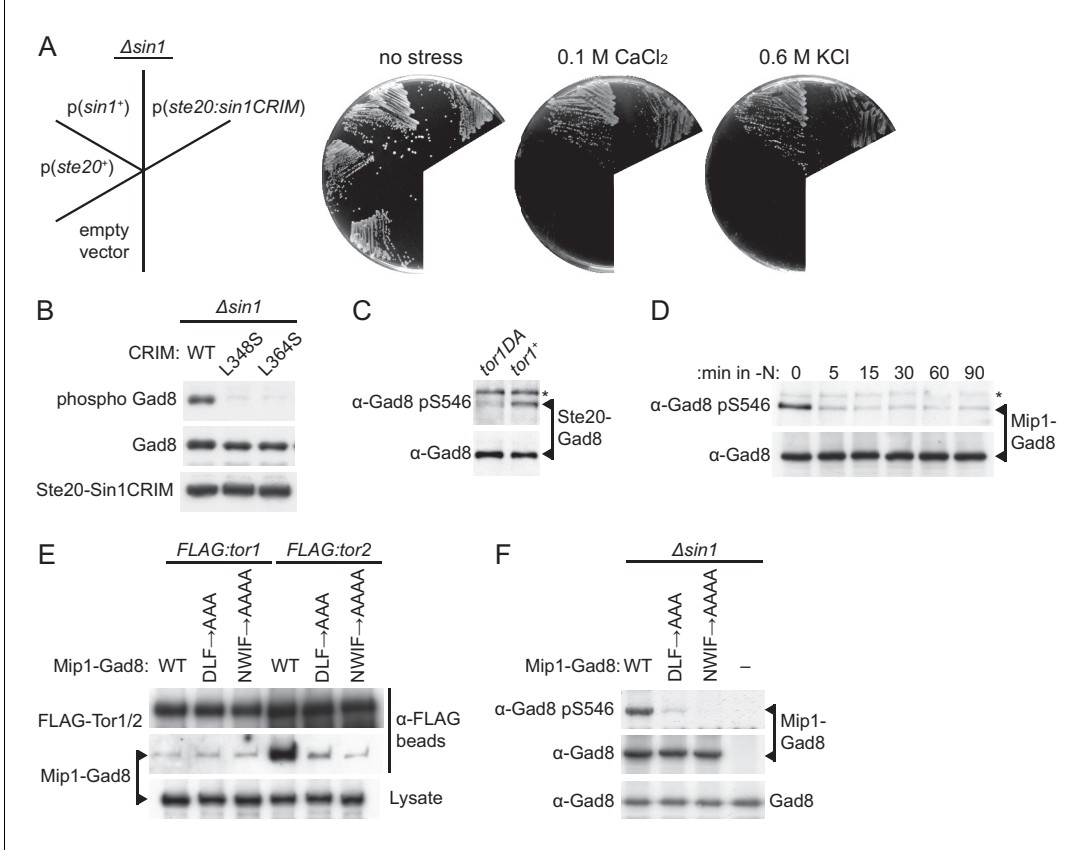

**Figure 5.** Substrate recruitment by Sin1CRIM determines the specificity of TORC2. (**A** and **B**) SpSin1CRIM fused to the Ste20 subunit can substitute for the Sin1 function in TORC2. In (**A**), a Δ*sin1* strain (CA5126) was transformed with an empty vector, or a plasmid to express the *ste20⁺*, *sin1⁺* or *ste20:sin1CRIM* genes. Growth of the transformants was tested at 30°C on YES plates with and without 0.1 M CaCl2 or 0.6 M KCl. In (**B**), phosphorylation of Gad8-S546 was examined by immunoblotting in a Δ*sin1* strain (CA5126) carrying a plasmid to express FLAG-tagged Ste20 fused to SpSin1CRIM with and without L348S/L364S mutations. (**C**) TORC2 without Sin1 can phosphorylate Gad8 fused to the Ste20 subunit. Δ*sin1* strains with *tor1⁺* (CA7471) and *tor1-D2137A* (CA7395) alleles were transformed with a plasmid to express Ste20 fused to Gad8. Expression and phosphorylation of the Ste20-Gad8 fusion were detected by antibodies against Gad8 and phosphorylated Gad8-S546, respectively. (**D**) Gad8 fused to the Mip1 subunit of TORC1 is phosphorylated when TORC1 is active. A Δ*sin1* strain (CA5126) carrying a plasmid to express Gad8 fused to Mip1, the Raptor ortholog in fission yeast, was grown to mid-log phase, followed by nitrogen starvation (-N) to inactivate TORC1. Expression and phosphorylation of the Mip1-Gad8 fusion were analyzed as in (**C**). (**E**) The Mip1-Gad8 fusion is incorporated into TORC1. Mip1-Gad8 was expressed in *FLAG:tor1* (CA6530) and *FLAG:tor2* (CA6904) strains, from which TORC2 and TORC1 were collected onto anti-FLAG beads, respectively, and co-purification of Mip1-Gad8 was examined by anti-Gad8 immunoblotting. Alanine substitutions of Mip1 residues 316–318 ('DLF') or 366–369 ('NWIF') prevented the fusion protein from being incorporated into TORC1. (**F**) Phosphorylation of Gad8-S546 in the Mip1-Gad8 fusion protein incorporated into TORC1. A Δ*sin1* strain (CA5126) was transformed with an empty vector (−) or plasmids to express Mip1-Gad8 with or without the mutations that abrogate its incorporation into TORC1 as shown in (**E**). Expression and phosphorylation of the Mip1-Gad8 proteins were analyzed as in (**D**). The endogenous Gad8 was monitored as a loading control (bottom).

The following figure supplements are available for figure 5:

**Figure supplement 1.** The CRIM domain alone is not sufficient to replace the Sin1 function in vivo.

**Figure supplement 2.** When fused to Ste20, SpSin1CRIM can replace the Sin1 function in a manner dependent on its ability to bind Gad8.

hybrid assays (*Figure 2—figure supplement 4*), implying the evolutionarily conserved structure and function of CRIM. Budding yeast TORC2 additionally requires a PH domain protein, Slm1, to recruit the Ypk1 kinase for phosphorylation (*Berchtold et al., 2012*; *Niles et al., 2012*). However, the unique Slm ortholog SPAC637.13c in fission yeast has no apparent role in Gad8 phosphorylation by TORC2 (manuscript in preparation), and no Slm ortholog has been found in metazoan.

CRIM is the signature sequence of Sin1 orthologs in diverse eukaryotes (*Schroder et al., 2004*), and its solution structure shows a ubiquitin-like fold. Ubiquitin superfold domains identified in intracellular signaling proteins often mediate protein-protein interaction; their interaction surfaces are diverse, involving different segments of the ubiquitin fold as well as loop regions (*Kiel and Serrano, 2006*; *Sumimoto et al., 2007*). We found that SpSin1CRIM has a distinctive acidic loop that projects from the ubiquitin-like fold. Our experiments with human SIN1 also suggested a critical role of the conserved acidic stretch in the SIN1-AKT interaction and phosphorylation of AKT. Eliminating the sequence that spans this acidic stretch of SIN1 also compromises mTORC2-dependent phosphorylation of PKCα (*Cameron et al., 2011*), although such a relatively large deletion may disturb the overall folding of Sin1CRIM. It should be noted that our results do not exclude the possibility of additional SIN1-substrate interaction outside of the CRIM domain. Like AKT, SGK1 binds HsSin1-CRIM (*Figure 2D*), but it was also reported that the Q68H mutation to the N-terminal region of SIN1 impairs its interaction with SGK1 (*Lu et al., 2011*). The Q68H mutation, however, does not affect SIN1-AKT interaction and therefore, additional SIN1-substrate interaction outside of the CRIM domain may be substrate-specific.

Yeast two-hybrid analyses showed that HsSin1CRIM interacts with the kinase catalytic domain of AKT (*Figure 2—figure supplement 1D*). On the other hand, the kinase domain fragment of Gad8 lacking the N-terminal, non-catalytic domain failed to interact with SpSin1CRIM (*Figure 2—figure supplement 1B*). Interestingly, this N-terminal region of Gad8 and its budding yeast orthologs, Ypk1 and Ypk2, contain a conserved auto-inhibitory sequence (*Kamada et al., 2005*; our unpublished results); without the N-terminal auto-inhibitory domain, the Gad8 kinase domain is expected to be in its active conformation, which might affect the interaction with the SpSin1CRIM. We have not succeeded in localizing the CRIM interaction to any subfragment of the Gad8/AKT catalytic domains, and further study is required to elucidate how Sin1CRIM recognizes the TORC2 substrate kinases, including Sin1CRIM–kinase co-crystallization.

We show that Gad8, a TORC2 substrate, can be phosphorylated by TORC1 when recruited to TORC1 by gene fusion. Thus, the substrate specificity of the TOR complexes seems to be determined by their ability to recruit specific substrates through the regulatory subunits. In mTORC1, RAPTOR is such a regulatory subunit that recognizes the TOR Signaling (TOS) motif in S6K1 and 4E-BP1 (*Hara et al., 2002*; *Schalm and Blenis, 2002*; *Schalm et al., 2003*), although many TORC1 substrates have no apparent TOS motif. Our study has shown that Sin1 is the substrate-recruiting subunit of TORC2, and that its CRIM domain is the determinant of the substrate specificity. Hyperactivation of mTORC2 is often a key event in cancer development (*Guertin et al., 2009*; *Lee et al., 2012*; *Lawrence et al., 2014*; *Carr et al., 2015*), but there is no effective drug that specifically inhibits mTORC2. A possible strategy for development of mTORC2 inhibitors is to find small molecules that target the substrate-binding interface of mTORC2 (*Sparks and Guertin, 2010*), and our study is expected to serve as a basis for such an approach.

## Materials and methods

### General techniques with fission yeast

*S. pombe* strains used in this study are listed in *Table 2*. Growth media and basic techniques for *S. pombe* have been described previously (*Alfa et al., 1993*). Stress sensitivity of *S. pombe* strains was assessed by streaking or spotting on YES agar plates at 37°C or those containing 2 M sorbitol, 1 M KCl or 0.1 M CaCl$_2$ at 30°C (*Wang et al., 2005*). Gene disruption and gene tagging with the epitope sequences have been described previously (*Bähler et al., 1998*; *Shiozaki and Russell, 1997*). Quik-Change kit (Stratagene) and PrimeSTAR Mutagenesis Basal Kit (Takara Bio Inc) were used for site-directed mutagenesis.

### Yeast two-hybrid assay and screen

To isolate Gad8-interacting proteins by yeast two-hybrid assays, the entire open reading frame of *gad8+* was cloned in the pGBT9 vector (Clontech) to express the *GAL4* DNA-binding domain fused with Gad8. A fission yeast cDNA library constructed in the pGAD GH vector (Clontech) and the budding yeast HF7c strain were used (*Tatebe et al., 2008*). Yeast transformants were screened by histidine auxotrophy and β-galactosidase assay.

**Table 2.** *S. pombe* strains used in this study.

| Strain ID | Genotype | Source, Reference |
| --- | --- | --- |
| CA101 | *h⁻ leu1–32* | Lab stock |
| CA4593 | *h⁻ leu1–32 ura4-D18 Δtor1::ura4⁺* | (*Kawai et al., 2001*) |
| CA4767 | *h⁻ leu1–32 ura4-D18 Δtor1::ura4⁺ Δsin1::kanMX6* | (*Ikeda et al., 2008*) |
| CA4855 | *h⁻ leu1–32 ura4-D18 Δtor1::ura4⁺ ste20:5FLAG(kanR)* | This study |
| CA5021 | *h⁻ leu1–32 Δste20::kanMX6* | (*Ikeda et al., 2008*) |
| CA5126 | *h⁻ leu1–32 Δsin1::kanMX6* | (*Ikeda et al., 2008*) |
| CA5142 | *h⁻ leu1–32 ura4-D18 Δgad8::ura4⁺* | (*Matsuo et al., 2003*) |
| CA5827 | *h⁻ leu1–32 ura4-D18 Δtor1::ura4⁺ Δgad8::ura4⁺* | This study |
| CA6275 | *h⁻ leu1–32 ste20:5FLAG(kanR) Δsin1::kanMX6* | This study |
| CA6281 | *h⁻ leu1–32 gad8:5FLAG(kanR)* | (*Tatebe et al., 2010*) |
| CA6323 | *h⁻ leu1–32 tor1D2137A* | This study |
| CA6435 | *h⁻ leu1–32 ste20:13myc(kanR)* | (*Tatebe et al., 2010*) |
| CA6530 | *h⁻ leu1–32 FLAG:tor1(hph)* | (*Hayashi et al., 2007*) |
| CA6855 | *h⁻ leu1–32 FLAG:tor1(hph) bit61:13myc(kanMX6)* | (*Tatebe and Shiozaki, 2010*) |
| CA6859 | *h⁻ leu1–32 bit61:13myc(kanMX6)* | (*Tatebe and Shiozaki, 2010*) |
| CA6870 | *h⁻ leu1–32 FLAG:tor1(hph) Δsin1::kanMX6* | This study |
| CA6904 | *h⁻ leu1–32 FLAG:tor2(kanR)* | (*Hayashi et al., 2007*) |
| CA6984 | *h⁻ leu1–32 sin1:13myc(kanMX6)* | (*Tatebe et al., 2010*) |
| CA6993 | *h⁻ leu1–32 sin1:13myc(kanMX6) gad8:5FLAG(kanMX6)* | (*Tatebe et al., 2010*) |
| CA7087 | *h⁻ leu1–32 FLAG:tor1(hph) ste20:13myc(kanMX6)* | (*Hatano et al., 2015*) |
| CA7092 | *h⁻ leu1–32 FLAG:tor1(hph) sin1:13myc(kanMX6)* | (*Hatano et al., 2015*) |
| CA7143 | *h⁻ leu1–32 FLAG:tor1(hph) ste20:13myc(kanMX6) Δsin1::kanMX6* | This study |
| CA7147 | *h⁻ leu1–32 FLAG:tor1(hph) sin1:13myc(kanMX6) Δgad8::ura4⁺* | This study |
| CA7150 | *h⁻ leu1–32 FLAG:tor1(hph) sin1:13myc(kanMX6) Δbit61::ura4⁺* | This study |
| CA7151 | *h⁻ leu1–32 FLAG:tor1(hph) ste20:13myc(kanMX6) Δwat1::kanMX6* | This study |
| CA7155 | *h⁻ leu1–32 FLAG:tor1(hph) ste20:13myc(kanMX6) Δbit61::ura4⁺* | This study |
| CA7172 | *h⁻ leu1–32 FLAG:tor1(hph) Δsin1::kanMX6 bit61:13myc(kanMX6)* | This study |
| CA7183 | *h⁻ leu1–32 wat1:13myc(kanMX6)* | This study |
| CA7189 | *h⁻ leu1–32 ura4-D18 FLAG:tor1(hph) ste20:13myc(kanMX6) Δgad8::ura4⁺* | This study |
| CA7200 | *h⁻ leu1–32 FLAG:tor1(hph) sin1:13myc(kanMX6) Δwat1::kanMX6* | This study |
| CA7213 | *h⁻ leu1–32 FLAG:tor1(hph) wat1:13myc(kanMX6)* | (*Hatano et al., 2015*) |
| CA7217 | *h⁻ leu1–32 FLAG:tor1(hph) Δste20::kanMX6 bit61:13myc(kanMX6)* | This study |
| CA7222 | *h⁺ leu1–32 ura4-D18 his7–366 FLAG:tor1(hph) Δste20::kanMX6 sin1:13myc(kanMX6)* | This study |
| CA7286 | *h⁻ leu1–32 FLAG:tor1(hph) Δsin1::kanMX6 wat1:13myc(kanMX6)* | This study |
| CA7307 | *h⁻ leu1–32 ura4-D18 FLAG:tor1(hph) bit61:13myc(kanMX6) Δgad8::ura4⁺* | This study |
| CA7317 | *h⁻ leu1–32 ura4-D18 FLAG:tor1(hph) wat1:13myc(kanMX6) Δbit61::ura4⁺* | This study |
| CA7318 | *h⁻ leu1–32 ura4-D18 FLAG:tor1(hph) wat1:13myc(kanMX6) Δgad8::ura4⁺* | This study |
| CA7319 | *h⁻ leu1–32 FLAG:tor1(hph) Δwat1::kanMX6 bit61:13myc(kanMX6)* | This study |
| CA7329 | *h⁻ leu1–32 ura4-D18 FLAG:tor1(hph) Δste20::kanMX6 wat1:13myc(kanMX6)* | This study |
| CA7395 | *h⁻ leu1–32 (hph)FLAG:tor1D2137A(kanMX6) Δsin1::kanMX6* | This study |
| CA7471 | *h⁻ leu1–32 Δste20::kanMX6 Δsin1::kanMX6* | This study |
| CA8070 | *h⁻ leu1–32 psk1:5FLAG(kanMX6)* | This study |

Yeast two-hybrid screens to isolate Sin1CRIM mutants defective in the interaction with Gad8 was conducted as follows. Randomly mutagenized DNA fragments encoding the fission yeast Sin1CRIM were prepared by standard PCR using premix Ex Taq (Takara Bio Inc.). pGAD GH was linearized by restriction enzyme digestion at two sites in the multi cloning site. The budding yeast HF7c strain carrying pGBT9-*gad8*<sup></sup> was transformed with both mutagenized SpSin1CRIM DNA fragments and linearized pGAD GH. Resultant transformants were screened by histidine auxotrophy. GAL4 transcriptional activation domain-SpSin1CRIM fusion in each candidate transformant was examined in immunoblotting with antibodies against GAL4 transcriptional activation domain (RRID:AB_669111) to eliminate all the nonsense mutations. Deficiency in the interaction was verified by recovery and re-transformation of pGAD GH-mutant SpSin1CRIM. Mutation sites were determined by DNA sequencing.

## Preparation of recombinant protein expressed in *E. coli*

pGEX-KG or pMAL-c2x expression plasmids carrying various Sin1 fragments were transformed into BL21 *E. coli* strain. Exponentially growing BL21 cultures were transferred from 37°C to 16°C, followed by IPTG addition at the final concentration of 0.1 mM. Cultures were incubated overnight at 16°C for maximum production of recombinant protein. Cells were harvested by centrifugation and stored at −80°C until use. Frozen cells were thawed in ice-cold TBS (20 mM Tris-HCl [pH 7.5], 150 mM NaCl) and disrupted by sonication. After adding Triton X-100 at the final concentration of 1%, lysate was cleared by 15-min centrifugation at 20,800 x g. Supernatant was incubated with glutathione sepharose (GE Healthcare) or amylose resin (New England Biolabs) for 1 hr, followed by extensive wash with ice-cold TBS containing 1% Triton X-100. Cell breakage and all the following purification steps were performed at 4°C.

## Human cell culture

Human HEK-293T (RRID:CVCL_0063) and MCF-7 (RRID:CVCL_0031) cells were cultured in DMEM (Wako) supplemented with 10% (v/v) FBS. HEK-293T (RCB2202) and MCF-7 (RCB1904) were obtained from RCB (Riken Cell Bank, Japan), neither of which we further authenticated. Mycoplasma contamination was examined by DAPI staining and fluorescence microscopy. Transfection was performed with Lipofectamine 2000 (Life Technologies Japan Ltd.) or polyethylenimine. FLAG-tagged AKT2, PKC, SGK1, and S6K1 are transiently expressed under the control of the CMV promoter of p3xFLAG-CMV-7.1 (Sigma-Aldrich).

## Generation of *SIN1*-deficient human culture cells by the CRISPR/Cas9 system

The guide sequence 'GGACACCGATTTCCCCCCGC' targeting Exon 6 of the human *SIN1* gene was cloned into pX330 (Addgene #42230) (*Cong et al., 2013*). pCAG-EGxxFP was used to examine efficiency of the target DNA cleavage by the guide sequence and Cas9 (*Mashiko et al., 2013*). The resultant plasmid DNA was co-transfected with the puromycin-resistant marker DNA into MCF-7 human culture cells. Forty-eight hours after co-transfection, cells were re-plated in growth media containing 0.4 µg/ml puromycin, followed by isolation of drug-resistant colonies. Expression of full length SIN1 protein in each isolated clone was tested by immunoblotting with anti-SIN1 antibodies (RRID:AB_661901) (*Figure 4—figure supplement 6A*). Genomic DNA flanking the guide and PAM sequence in SIN1 was amplified by PCR and cloned for DNA sequencing (*Figure 4—figure supplement 6B*).

## Protein-protein interactions

Protein-protein interactions were tested by co-precipitation experiments followed by immunoblot detection. Luminescent Image Analyzer LAS-4000 (Fujifilm) was used for quantification in immunoblot.

Fission yeast cell lysate was prepared in lysis buffer (20 mM HEPES-KOH [pH 7.5], 150 mM potassium glutamate or sodium glutamate, 0.25% Tween-20, 50 mM NaF, 10 mM sodium pyrophosphate, 10 mM p-nitrophenyl phosphate, 10 mM β-glycerophosphate, PMSF, aprotinin, leupeptin and the protease inhibitor cocktail for use in purification of Histidine-tagged proteins [Sigma-Aldrich P8849]), followed by centrifugation for 15 min at 20,800 x g. The total protein levels of supernatant

were quantified by Bradford assay. For interaction between myc-tagged protein and FLAG-tagged protein in fission yeast cells, supernatant was prepared and incubated for 2 hr with anti-FLAG M2-affinity gel (RRID:AB_10063035) or EZview Red anti-c-myc-affinity gel (RRID:AB_10093201), followed by extensive wash. Resultant samples were subjected to immunoblotting with anti-c-*myc* (RRID:AB_ 631274; RRID:AB_10092917) and anti-FLAG (RRID:AB_10596509; RRID:AB_259529) antibodies. For interaction of Gad8FLAG with GST-SpSin1CRIM expressed in fission yeast, GST-SpSin1CRIM expression was induced under the control of the thiamine-repressive *nmt1* promoter in a fission yeast *gad8:FLAG* strain. Supernatant was prepared and incubated with glutathione sepharose (GE Healthcare) for 2 hr, followed by extensive wash. GST-SpSin1CRIM and Gad8-FLAG were detected in Coomassie Brilliant Blue (CBB) staining and immunoblotting with anti-FLAG M2 antibodies (RRID:AB_ 10596509; RRID:AB_259529), respectively. For interaction of Gad8-FLAG with bacterially produced GST-SpSin1CRIM, GST-SpSin1CRIM was first purified onto glutathione sepharose. GST-SpSin1CRIM bound to glutathione sepharose was incubated for 1 hr with supernatant prepared from a fission yeast *gad8:FLAG* strain. After extensive wash, precipitates were subjected to CBB staining for GST-SpSin1CRIM detection and immunoblotting with anti-FLAG antibodies (RRID:AB_10596509; RRID: AB_259529) for Gad8-FLAG detection, respectively.

For interaction of human AGC kinases with HsSin1CRIM, human cell lysate was prepared in lysis buffer (20 mM HEPES-KOH [pH 7.5], 150 mM sodium glutamate, 10% glycerol, 0.25% tween-20, PMSF, and the protease inhibitor cocktail for use in purification of Histidine-tagged proteins [Sigma P8849]). Bacterially expressed GST-HsSin1CRIM and MBP-HsSin1CRIM were purified onto glutathione sepharose (GE Healthcare) and amylose resin (New England Biolabs), respectively, followed by incubation with human cell lysate.

## NMR study

The experimental procedure for structure determination of SpSin1CRIM with MTSL spin labels has been reported previously (*Furuita et al., 2014*; *Kataoka et al., 2014*). In brief, the cDNA encoding SpSin1CRIM was inserted into pCold-GST vector (*Hayashi and Kojima, 2008*). Nine single cysteine mutants were generated using the QuikChange site-directed mutagenesis method (Stratagene). The wild-type and mutant proteins were overexpressed in Escherichia coli RosettaTM (DE3) (Novagen). Following purification, the spin-labeled reagent MTSL [(1-oxyl-2,2,5,5-tetramethyl-Δ3-pyrroline-3-methyl) methanethiosulfonate] was attached to the thiol moiety of the introduced cysteine residues of the mutant proteins. Chemical shift assignments were performed using a conventional method. Dihedral angles were estimated using the program TALOS+ (*Shen et al., 2009*) and three bond JC' Cγ and JNCγ coupling constants (*Hu and Bax, 1997*). In order to obtain NOE distance restraints, 15N- and 13C-edited NOESY spectra of the wild-type protein were recorded. PRE distance restraints were calculated using intensity ratios of 1H-15N HSQC spectra of MTSL-conjugated mutant proteins in the paramagnetic and diamagnetic states (*Battiste and Wagner, 2000*). The structure calculations and automated NOE assignments were performed by CYANA 3.95 (*Güntert et al., 1997*), in which PRE distance restraints were used in combination. The obtained structures were refined using Xplor-NIH 2.31 (*Schwieters et al., 2006*, *2003*) with a single MTSL nitroxide label at each mutated position.

The 10 MTSL-conjugated structures (PDB ID: 2RUJ) were further refined using Xplor-NIH 2.31. First, MTSL-conjugated cysteines in each structure were mutated back to wild-type residues, and each structure was subjected to energy minimization. Then, using these structures as initial models, structure refinements were performed with NOE distance, PRE distance and dihedral angle restraints. The PRE distance restraints were introduced between Cβ of mutated residues and amide protons with the error of ±7 Å. A total of 100 structures (10 structures per initial model) were calculated, and the 10 lowest energy structures were selected and analyzed. The structural restraints and statistics are summarized in *Table 1*.

## Data availability

The atomic coordinates of the refined SpSin1CRIM structure have been deposited in the Protein Data Bank with accession code 2RVK.

## Acknowledgements

We are grateful to N Tahara and B You for technical assistance. We also thank M Ikawa, M Yamamoto, M Yanagida and the National BioResource Yeast Project of Japan for strains and plasmids, and M Saito for technical advice. This work was supported by grants from NIH, Mitsubishi Foundation, Daiichi Sankyo Foundation of Life Science, and Astellas Foundation for Research on Metabolic Disorders to KS, JSPS Grants-in-Aid to KS, HT and CK, grants from Foundation for NAIST and Sagawa Foundation for Promotion of Cancer Research to HT, and by the Platform Project for Supporting in Drug Discovery and Life Science Research from the MEXT and AMED, Japan. TH was a recipient of JSPS Research Fellowship for Young Scientists.

## Additional information

### Funding

| Funder | Grant reference number | Author |
|---|---|---|
| National Institute of General Medical Sciences | GM059788 | Kazuhiro Shiozaki |
| Japan Society for the Promotion of Science | 23870019 | Kazuhiro Shiozaki |
| Ministry of Education, Culture, Sports, Science, and Technology | | Kazuhiro Shiozaki |
| Japan Agency for Medical Research and Development | | Kazuhiro Shiozaki |
| Mitsubishi Foundation | | Kazuhiro Shiozaki |
| Daiichi Sankyo Foundation of Life Science | | Kazuhiro Shiozaki |
| Astellas Foundation for Research on Metabolic Disorders | | Kazuhiro Shiozaki |
| Sagawa Foundation for Promotion of Cancer Research | | Hisashi Tatebe |
| Foundation for NAIST | | Hisashi Tatebe |
| Japan Society for the Promotion of Science | 26291024 | Kazuhiro Shiozaki |
| Japan Society for the Promotion of Science | 16K14681 | Chojiro Kojima |
| Japan Society for the Promotion of Science | 26291013 | Chojiro Kojima |
| Japan Society for the Promotion of Science | 23770230 | Hisashi Tatebe |
| Japan Society for the Promotion of Science | 25440086 | Hisashi Tatebe |

The funders had no role in study design, data collection and interpretation, or the decision to submit the work for publication.

### Author contributions

HT, Identified the SpSin1CRIM-Gad8 interaction, Characterized the interaction in detail, Designed experiments, Interpreted results, Wrote the manuscript; SM, TH, Characterized the SpSin1CRIM-Gad8 interaction in detail, Assisted with preparing the manuscript; TY, Analyzed the interaction between HsSin1CRIM and the mTORC2 substrates, Assisted with preparing the manuscript; DR, Identified the SpSin1CRIM-Gad8 interaction, Assisted with preparing the manuscript; TF, Constructed the SIN1-defective MCF-7 cell line, Assisted with preparing the manuscript; SK, KF, CK, Determined the NMR structure of SpSin1CRIM, Assisted with preparing the manuscript; KS, Designed experiments, Interpreted results, Wrote the manuscript

Author ORCIDs

Kazuhiro Shiozaki, http://orcid.org/0000-0002-0395-5457

## Additional files

### Major datasets

The following dataset was generated:

| Author(s) | Year | Dataset title | Dataset URL | Database, license, and accessibility information |
|---|---|---|---|---|
| Furuita K, Kataoka S, Shiozaki K, Koji-ma C | 2017 | Refined solution structure of Schizosaccharomyces pombe Sin1 CRIM domain | http://pdbj.org/mine/summary/2rvk | Publicly available at the Protein Data Bank Japan (accession no: 2RVK). |

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
