## [Decision Letter]

Thank you for submitting your article "Substrate specificity of TOR complex 2 is determined by a ubiquitin-fold domain of the Sin1 subunit" for consideration by *eLife*. Your article has been reviewed by three peer reviewers, and the evaluation has been overseen by a Reviewing Editor and Tony Hunter as the Senior Editor. The reviewers have opted to remain anonymous.

The reviewers have discussed the reviews with one another and the Reviewing Editor has drafted this decision to help you prepare a revised submission.

Summary:

The authors show that Sin1 in *S. pombe*, unlike mammals, is not required for the catalytic activity of TORC2. They further demonstrate that the Conserved Region In the Middle (Sin1CRIM) forms a unique domain that specifically binds the substrate Gad8. They solved the 3-D structure of Sin1CRIM, which exhibits a ubiquitin-like fold with a characteristic acidic loop. The specific substrate recognition function is conserved in human Sin1CRIM. Importantly, the latter binds Akt.

Essential revisions:

The paper presents very strong data to support the authors' conclusions and clarifies discrepancies in the literature. Most of the revisions that are requested by the reviewers deal with explanations as to why certain experiments were not performed, and some control experiments that can be dealt with in a relatively short time. One comment that is of particular importance relates to mutations in the negatively charged loop that are drastic and might well disturb overall folding. Interactions of these residues in the structure should be shown. The reviewer asks whether milder variants have been tested? NMR would be suited to address this concern.

*Reviewer #1:*

The two complexes of TOR kinase, TORC1 & 2 are central regulators of cell growth, proliferation and dynamics. Tatebe et al. provide novel functional and structural insights to the role of Sin1 in TORC2 substrate recognition. They conclude that Sin1 in *S. pombe* (in contrast to mammals) is not required for TORC2 integrity and show that the CRIM rather than the PH/RBD domain of Sin1 is essential for TORC2-mediated phosphorylation of Gad8. A specific substrate recognition was also demonstrated for the CRIM domain of human Sin1. In a random mutagenesis screen the authors identify a series of mutations in S*. pombe* Sin1, which abolish Gad8, but not Tor1 interaction. The authors then present an NMR structure of *S. pombe* mSin1-CRIM, which reveals that the mutations identified in the mutagenesis screen affect buried residues and thus rather disturb the fold than the recognition mechanisms. In the structure, the authors identify a negatively charged loop as a potential interaction element, and vary this loop by mutating either all hydrophobic or all negatively charged residues; these drastic variations indeed diminish Gad8 interaction.

The role of Sin1-CRIM in substrate recognition is further demonstrated by fusing it to the Rictor homologue Ste20 in a Δsin1 background resulting in productive phosphorylation of Gad8 by TORC2. The fact that binding of Gad8 is sufficient for phosphorylation is indicated by the fact that Gad8 is phosphorylated even by TORC1 when fused to the Raptor homologue mip1.

The manuscript is well written and provides important novel insights to the role of Sin1 in TORC2 substrate recognition. The integration between the different parts of the manuscripts, in particular functional and structural studies, could be improved.

1) Results of random mutagenesis: The identified residues should be labelled in Figure 3—figure supplement 1, and implications of perturbed folding for different modes of TORC2 interaction should be discussed.

2) NMR studies: The NMR studies carried out by the authors would be optimally suited to monitor the folding state of all SIN1-CRIM variants used here and the authors should discuss why corresponding experiments have not been carried out.

3) The mutations in the negatively charged loop are drastic and might well disturb overall folding. Interactions of these residues in the structure should be shown. Have milder variants been tested? NMR would be suited to address this concern.

4) The minimal fragment of Sin1 necessary for the interaction with Gad8 is aa. 281-400. Further truncation from either end abolishes Gad8 interaction. The two flanking regions are distant in the folded domain. The authors should discuss whether this indicates multiple interactions sites.

5) In later experiments, the authors again used mutations L348S and L364S (Figure 3), which presumably affect the fold of Sin1-CRIM. Why weren't the variants in the acidic loop tested here?

*Reviewer #2:*

Tatebe et al. found that the TORC2 component, SIN1, in fission yeast is not required for the integrity of TORC2, but determines the substrate specificity of TORC2. They identified the conserved CRIM domain in SIN1 as the region that is required for binding to the substrate Gad8. Using truncation and point mutants, they also showed that human SIN1 could perform a similar function towards its substrate Akt. Structural analysis of the CRIM domain further identified an acidic loop that is essential for interaction with the TORC2 substrates. Fusion of Gad8 with Ste20, in the absence of SIN1, enabled Gad8 phosphorylation, suggesting that SIN1 is dispensable for TORC2 catalytic activity.

Using a combination of biochemical, genetic, and structural analyses, the study provides new insights on how TORC2 phosphorylates its substrates and elucidates the importance of SIN1. The results are quite robust and convincing and the fusion studies are particularly interesting. Some comments below could further support the conclusion and extend relevance to mammalian TORC2.

1) Do the truncation mutants in Figure 1 still bind TORC2? They should also test some of their mutants for binding to mammalian mTOR and rictor.

2) In Figure 2, binding of GAD8 to the CRIM-containing mutants should be compared with the RBD/PH-only containing mutants as negative control.

3. Phosphorylation of other mTORC2 target sites (e.g. Akt T450) and substrates (e.g. PKC) should be examined in Figure 4. Do the mutants also bind mTOR and rictor?

4) Where was the analysis on sin1-CRM alone on Figure 5? This was stated in the text but not included in the figure. The effect of ste20-sin1CRM on growth upon stress was also rather weak yet Gad8 was strongly phosphorylated by this mutant. Could this mutant be less efficient for phosphorylation of other AGC targets of TORC2?

*Reviewer #3:*

The mTORC2 kinase plays an important role in conveying growth factor signals to downstream AGC family protein kinases by phosphorylating their hydrophobic motif, an essential step in their activation. Unlike the related mTORC1 kinase, for which a substrate recruitment motif named TOS has been established, how mTORC2 recruits its substrates has not been well understood. While some studies have implicated the mTORC2 subunit Sin1 in substrate recruitment, other studies have reached contradictory conclusions.

In this manuscript, Tatebe et al. provide a wealth of experimental data conclusively showing that Sin1 recruits the Gad8 kinase and its human homologue AKT, and that this interaction is necessary for substrate phosphorylation. The data include yeast two hybrid assays, IPs and mutagenesis, which map the interaction to the Sin1 CRIM domain, as well as heterologous fusion of the CRIM domain to other mTORC2 subunits. They also report the solution structure of the CRIM domain, which adopts a ubiquitin fold and has an acidic loop to which most of the mutations that abrogate Gad8 binding map. The authors then go on to identify similar interactions between human Sin1 CRIM and AKT. Overall this manuscript is well written and the experiments are well executed with proper controls. The data supports the authors' conclusions.

The only drawback is that it has been previously reported (Cameron et al. 2011) that human SIN1 interacts with the mTORC2 substrates PKC epsilon and AKT through essentially the same CRIM acidic patch, although I agree with the authors that this has been somewhat controversial in the field (discussed by the authors in the second paragraph of the subsection “Sin1 CRIM domain binds the AGC kinases that are phosphorylated by TORC2”). As it settles this question, and also provides detailed structural and mutational data, I believe the manuscript merits publication in *eLife*.

---

## [Author Response]

*Essential revisions:*

*The paper presents very strong data to support the authors' conclusions and clarifies discrepancies in the literature. Most of the revisions that are requested by the reviewers deal with explanations as to why certain experiments were not performed, and some control experiments that can be dealt with in a relatively short time. One comment that is of particular importance relates to mutations in the negatively charged loop that are drastic and might well disturb overall folding. Interactions of these residues in the structure should be shown. The reviewer asks whether milder variants have been tested? NMR would be suited to address this concern.*

In the “poly NQ” mutant that we used in Figure 4, the multiple acidic residues in the loop region were substituted with uncharged Asn or Gln. We predicted that those substitutions would not disturb the overall folding of the CRIM domain, because the side chains of those acidic residues in this region face outward and do not interact with other residues outside of the loop (new Figure 4—figure supplement 1). As suggested, we expressed and purified the Sin1CRIM with the “poly NQ” mutation for NMR analysis; the mutant protein has been found very stable, and its NMR spectrum resembles that of the wild-type Sin1CRIM, indicating that the overall folding of Sin1CRIM is not disturbed by the “poly NQ” mutation (new Figure 4—figure supplement 3). We have also tested amino acid substitutions of the individual acidic residues within the loop region, showing that the acidic residues in the second half of the loop (D358, E359 and D360) play particularly important roles in the CRIM-Gad8 interaction (new Figure 4—figure supplement 4). We are grateful to reviewer #1 and the Editors for the helpful suggestions that have led to better characterization of the loop region.

Reviewer #1:

*[…] The manuscript is well written and provides important novel insights to the role of Sin1 in TORC2 substrate recognition. The integration between the different parts of the manuscripts, in particular functional and structural studies, could be improved.*

*1) Results of random mutagenesis: The identified residues should be labelled in Figure 3—figure supplement 1, and implications of perturbed folding for different modes of TORC2 interaction should be discussed.*

The residues identified through the random mutagenesis are labeled in Figure 3—figure supplement 1. Implications of the folding perturbation by these mutations are discussed in the third paragraph of the subsection “Sin1 CRIM has a protruding acidic loop involved in the recruitment of TORC2 substrates”.

*2) NMR studies: The NMR studies carried out by the authors would be optimally suited to monitor the folding state of all SIN1-CRIM variants used here and the authors should discuss why corresponding experiments have not been carried out.*

We have now included the NMR analysis of the poly NQ mutant of Sin1CRIM as new Figure 4—figure supplement 3. We did not pursue the structures of the other inactive mutant proteins in this study, because the current report focuses on the unequivocal identification of the CRIM domain as the substrate recognition domain of TORC2. Further structural biological and biophysical studies of the CRIM folding will be conducted in the near future.

*3) The mutations in the negatively charged loop are drastic and might well disturb overall folding. Interactions of these residues in the structure should be shown. Have milder variants been tested? NMR would be suited to address this concern.*

Please see response to Essential revisions above.

*4) The minimal fragment of Sin1 necessary for the interaction with Gad8 is aa. 281-400. Further truncation from either end abolishes Gad8 interaction. The two flanking regions are distant in the folded domain. The authors should discuss whether this indicates multiple interactions sites.*

As shown in Figure 2—figure supplement 1, Sin1 aa. 281-400, but not the N-terminally truncated 301-400 fragment, can interact with Gad8. This 301-400 fragment lacks both α1 helix and β1 sheet and therefore, it is very likely that the overall structure of the CRIM domain is significantly disturbed.

The Figure 2 experiment found that Sin1 aa. 281-400, but not the C-terminally truncated 281-380 fragment, can interact with Gad8. This 281-381 fragment lacks α3 helix, which is integral part of the CRIM folding.

Thus, further truncation of Sin1 aa. 281-400 from either end abolishes Gad8 interaction, most likely because those truncations lead to disruption of the overall CRIM structure. Therefore, it is difficult to conclude that the truncation data indicates multiple interaction sites within Sin1CRIM.

*5) In later experiments, the authors again used mutations L348S and L364S (Figure 3), which presumably affect the fold of Sin1-CRIM. Why weren't the variants in the acidic loop tested here?*

It is not clear which experiments are referred to as “later experiments” here, but we assume that the reviewer meant the Figure 5 experiment. Figure 5 demonstrates that Sin1CRIM fused to Ste20 can complement the *∆sin1* defect in Gad8 phosphorylation and that the complementation ability of this fusion protein is dependent on the functional CRIM domain. Thus, any mutation that abrogates the CRIM function serves as negative control in this particular experiment, including L348S and L364S. To further confirm the conclusion of these experiments, we have now included new Figure 5—figure supplement 2, which shows that the “poly NQ” and “AA” variants in the acidic loop fail to function even when fused to Ste20.

Reviewer #2:

*[…] Using a combination of biochemical, genetic, and structural analyses, the study provides new insights on how TORC2 phosphorylates its substrates and elucidates the importance of SIN1. The results are quite robust and convincing and the fusion studies are particularly interesting. Some comments below could further support the conclusion and extend relevance to mammalian TORC2.*

*1) Do the truncation mutants in Figure 1 still bind TORC2? They should also test some of their mutants for binding to mammalian mTOR and rictor.*

New Figure 1—figure supplement 2 has been included to show that the Sin1 truncation mutants 1-290 and ∆291-390 in Figure 1 still bind Tor1 and Ste20 (the rictor ortholog in *S. pombe*) to form TORC2, though they are not functional in vivo (Figure 1). These results are consistent with the specific role of CRIM in the interaction with Gad8.

We have also tested four different mutants of human Sin1 for their interaction with mTOR and rictor, showing as new Figure 4—figure supplement 7; it has been confirmed that the CRIM domain is not required for the interaction of human Sin1 with mTOR and rictor.

*2) In Figure 2, binding of GAD8 to the CRIM-containing mutants should be compared with the RBD/PH-only containing mutants as negative control.*

The rationale of this additional experiment is not clear, because the Figure 2 experiment already has negative controls of GST-Sin1“281-380 aa” and “247-380 aa”. Please note that the yeast 2-hybrid assay shown in Figure 2—figure supplement 1 showed that the Sin1 C-terminal fragment containing the RBD and PH domains (329-665 aa) cannot bind Gad8.

*3. Phosphorylation of other mTORC2 target sites (e.g. Akt T450) and substrates (e.g. PKC) should be examined in Figure 4. Do the mutants also bind mTOR and rictor?*

Phosphorylation of Akt-T450 and PKCα-S657 have been examined as suggested and shown as new Figure 4—figure supplement 8; not surprisingly, phosphorylation of those TORC2 target sites is also compromised in cells expressing Sin1 with mutations to its CRIM domain. New Figure 4—figure supplement 7 shows that the CRIM domain mutants of human Sin1 bind mTOR and rictor, confirming that the CRIM domain is not required for interaction with mTOR and rictor.

*4) Where was the analysis on sin1-CRM alone on Figure 5? This was stated in the text but not included in the figure. The effect of ste20-sin1CRM on growth upon stress was also rather weak yet Gad8 was strongly phosphorylated by this mutant. Could this mutant be less efficient for phosphorylation of other AGC targets of TORC2?*

The analysis on the in vivo functionality of Sin1CRIM alone is now included as Figure 5—figure supplement 1. In fission yeast, Gad8 is the only known AGC target of TORC2, and the stress-sensitive phenotypes of the *∆sin1* and *∆gad8* mutants are indistinguishable (Tatebe et al., 2010). Thus, it is more likely that TORC2 with Ste20-Sin1CRIM fusion is not fully functionally in phosphorylating Gad8, comparing to wild-type TORC2.

*Reviewer #3:*

*[…] As it settles this question, and also provides detailed structural and mutational data, I believe the manuscript merits publication in eLife.*

We thank the reviewer for their positive comments.